# FROSTER: FROZEN CLIP IS A STRONG TEACHER FOR OPEN-VOCABULARY ACTION RECOGNITION

**Xiaohu Huang**[1]    **Hao Zhou**[2]    **Kun Yao**[2]    **Kai Han**[1†]

[1] Visual AI Lab, The University of Hong Kong
[2] Department of Computer Vision Technology (VIS), Baidu Inc.
huangxiaohu@connect.hku.hk, zhouh156@mail.ustc.edu.cn
yaokun01@baidu.com, kaihanx@hku.hk

## ABSTRACT

In this paper, we introduce **FROSTER**, an effective framework for open-vocabulary action recognition. The CLIP model has achieved remarkable success in a range of image-based tasks, benefiting from its strong generalization capability stemming from pretaining on massive image-text pairs. However, applying CLIP directly to the open-vocabulary action recognition task is challenging due to the absence of temporal information in CLIP's pretraining. Further, fine-tuning CLIP on action recognition datasets may lead to overfitting and hinder its generalizability, resulting in unsatisfactory results when dealing with unseen actions. To address these issues, FROSTER employs a residual feature distillation approach to ensure that CLIP retains its generalization capability while effectively adapting to the action recognition task. Specifically, the residual feature distillation treats the frozen CLIP model as a teacher to maintain the generalizability exhibited by the original CLIP and supervises the feature learning for the extraction of video-specific features to bridge the gap between images and videos. Meanwhile, it uses a residual sub-network for feature distillation to reach a balance between the two distinct objectives of learning generalizable and video-specific features. We extensively evaluate FROSTER on open-vocabulary action recognition benchmarks under both base-to-novel and cross-dataset settings. FROSTER consistently achieves state-of-the-art performance on all datasets across the board. Project page: https://visual-ai.github.io/froster.

## 1 INTRODUCTION

Open-vocabulary action recognition aims to recognize action categories that may not have been seen during training. The study of action recognition was dominated by pure vision based methods, such as (Ji et al., 2012; Carreira & Zisserman, 2017; Wang et al., 2016; Feichtenhofer et al., 2019; Lin et al., 2019), which assumes a closed-set setting where the models are trained and evaluated on a set of fixed and predefined action categories. Recently, the emergence of large-scale vision-language models, such as CLIP (Radford et al., 2021) and ALIGN (Jia et al., 2021), has enabled image classification on an open vocabulary of categories. Inspired by this success, attempts have been made to apply the CLIP model to action recognition by processing the video frame-by-frame, as CLIP was pretrained on image-text pairs. However, directly utilizing pretrained CLIP models yields sub-optimal performance, as these models lack access to video-text data during pretraining. To bridge the domain gap between images and videos, methods have been proposed to fine-tune the CLIP model (Wang et al., 2021; Liu et al., 2023; Rasheed et al., 2023) or utilize efficient adaption/prompting techniques (Chen et al., 2022a; Yang et al., 2023; Ju et al., 2022; Wasim et al., 2023; Ni et al., 2022) to extract video-specific knowledge. While these methods have achieved success in the closed-set setting, their performance is inferior when it comes to unseen categories (see Tab. 1).

In Fig. 1, we compare the frozen CLIP and three CLIP-adapted video models, namely Action CLIP (Wang et al., 2021), AIM (Yang et al., 2023) and ST-Adapter (Pan et al., 2022), in an open-vocabulary setting. These models are fine-tuned (except for the frozen CLIP) on Kinetics-400 (Carreira & Zisserman, 2017) but evaluated on UCF-101 (Soomro et al., 2012), HMDB-51 (Kuehne

---

[†]Corresponding author.

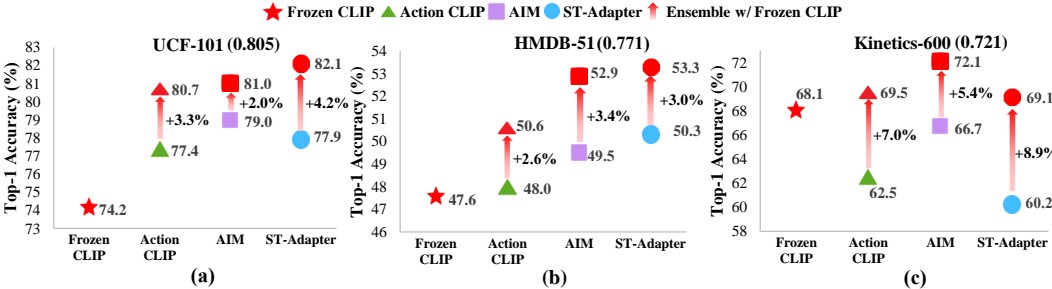

Figure 1: Performance comparison (Top-1 Acc (%)) under the open-vocabulary evaluation setting where the models are tuned on Kinetics-400, but evaluated on UCF-101, HMDB-51, and Kinetics-600. Note that shared categories between Kinetics-600 and Kinetics-400 are excluded when testing on Kinetics-600. The numbers in the brackets denote the semantic distance between training and evaluation datasets, which is measured by Hausdorff distance on text features of category names. Larger numbers denote higher similarities.

et al., 2011), and Kinetics-600 (Carreira et al., 2018) (*shared categories with Kinetics-400 are excluded*). Compared to the frozen CLIP, we observe that the tuned models achieve higher performance on UCF-101 and HMDB-51, but achieve inferior performance on Kinetics-600. This observation indicates that the generalization capacity of tuned models varies across different datasets. Further, using text features of category names, we measure the semantic distances between the training set (Kinetics-400) and testing sets (UCF-101, HMDB-51, and Kinetics-600). We find that, for testing sets that are more semantically similar to the training set (*i.e.*, UCF-101 and HMDB-51), the tuned models exhibit improvements over the frozen CLIP. However, when it comes to the semantically less similar dataset, namely Kinetics-600, which demands stronger generalizability, the performance of all tuned models deteriorates compared to the frozen CLIP. This suggests a decline in generalizability after fine-tuning. Based on these observations, we believe that a CLIP-based video model should possess the following properties: (1) The CLIP model should acquire video-specific knowledge to bridge the gap between images and videos, effectively addressing the challenges posed by the image-to-video domain transition. This entails adapting the model to understand and extract meaningful information from video data, taking into account temporal dynamics and context. (2) It is crucial to preserve the strong generalization capability of the pretrained CLIP model within the video model. This ensures that the video model can effectively generalize across different video datasets and tasks, leveraging the learned representations from the pretraining phase.

To validate the hypothesis made above, we evaluate the performance of each adapted model by ensembling it with the frozen CLIP by summing up their respective outputs. The results in Fig. 1 demonstrate that the ensemble models exhibit significant improvements across all datasets, indicating their efficacy as open-vocabulary classifiers. However, the naive approach of ensembling models results in notable additional computational costs due to inferring two models simultaneously. This limitation hinders practical applications in real-world scenarios. Therefore, it is imperative to explore methods for consolidating the ensemble model's knowledge into a single model, mitigating the computational burden.

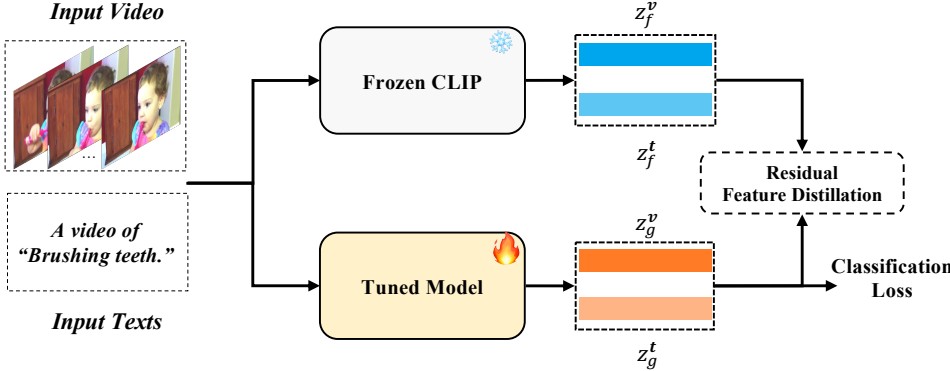

Figure 2: Overall idea of our FROSTER framework. It effectively learns feature representation that is both video-specific (via simple action-based fine-tuning) and generalizable (via our proposed residual feature distillation).

To this end, we propose FROSTER, a simple yet effective framework for open-vocabulary action recognition, which effectively learns feature representation that is both video-specific and generalizable. As shown in Fig. 2, 'video-specific' is achieved through common classification-based finetuning, while 'generalizable' is achieved by using frozen CLIP as a teacher to impart pretrained knowledge to the tuned model, inspired by knowledge distillation techniques (Hinton et al., 2015; Romero et al., 2014). The distillation process is akin to a regularization term that ensures the tuned features do not diverge too far from the frozen ones. Having two distinct objectives, we need to balance the feature learning between them. For instance, if we enforce the tuned features to be overly close to the frozen features, it may hinder the video-specific learning to fit the video data. Conversely, if we overemphasize video-specific learning, the generalizable capacity in the tuned model might be lost. To address this issue, we propose a residual feature distillation approach to balance feature learning between the two joint objectives.

Taking inspiration from the model patching method (Ilharco et al., 2022), a prior study (Weng et al., 2023) develops an open-vocabulary action recognition model through weight interpolation between the fine-tuned model and the frozen CLIP. This approach bears some resemblance of motivation to ours. Nevertheless, the weight interpolation technique necessitates the fine-tuned model and frozen CLIP to possess identical weight dimensions, thereby restricting its applicability across diverse network architectures (*e.g.*, adapter-based CLIP models). Also, methods (Zenke et al., 2017; Kirkpatrick et al., 2017) that penalize the updates of the pretrained layers are not applicable to the adapter-based methods since the pretrained layers are frozen.

We thoroughly evaluate the effectiveness of FROSTER on the two open-vocabulary settings, namely cross-dataset and base-to-novel, using the Kinectics-400 (Carreira & Zisserman, 2017), Kinetics-600 (Carreira et al., 2018), UCF-101 (Soomro et al., 2012), HMDB-51 (Kuehne et al., 2011), and Something-to-Something V2 (Goyal et al., 2017) datasets. The cross-dataset evaluation protocol tests models on a different dataset, while the base-to-novel evaluation protocol evaluates their ability to recognize unseen action categories within the same dataset. We couple FROSTER with different video recognition networks. In all cases, FROSTER demonstrates superior performance, showcasing its effectiveness and versatility.

The main contribution of this paper can be summarized as follows: Firstly, we introduce FROSTER, a simple yet effective framework for open-vocabulary action recognition. It can effectively learn feature representation that is both video-specific and generalizable. Secondly, we introduce a residual feature distillation approach to balance feature learning in both objectives. This technique mitigates potential conflicts and enables the model to achieve both goals simultaneously. Thirdly, we demonstrate the superiority of FROSTER through extensive evaluations on cross-dataset and base-to-novel settings across Kinetics-400, Kinetics-600, UCF-101, HMDB-51, and Something-to-Something V2 datasets with various network architectures.

## 2 RELATED WORK

### 2.1 CLIP-BASED ACTION RECOGNITION

Inspired by the strong representation of the pretrained CLIP model, many video recognition approaches have been proposed based on CLIP. These works can generally be categorized into two types: full fine-tuning (Wang et al., 2021; Liu et al., 2023; Rasheed et al., 2023) and partial fine-tuning (Chen et al., 2022a; Ni et al., 2022; Pan et al., 2022; Yang et al., 2023; Park et al., 2023). For full fine-tuning methods, ActionCLIP (Wang et al., 2021) is the first to introduce CLIP into video recognition, fine-tuning the CLIP model and applying additional temporal layers to model motion. Similarly, STAN (Liu et al., 2023) uses an auxiliary network to extract temporal features based on CLIP. ViFi-CLIP (Rasheed et al., 2023) shows that simply fine-tuning both the vision and the text encoders are also effective for action recognition. For partial fine-tuning, a common practice is to freeze the pretrained model parameters and introduce extra parameters for training. Inspired by the parameter-efficient fine-tuning in Natural Language Processing (Houlsby et al., 2019; Hu et al., 2021), Adaptformer (Chen et al., 2022a), X-CLIP (Ni et al., 2022), ST-Adapter (Pan et al., 2022), AIM (Yang et al., 2023), and DUALPATH (Park et al., 2023) employ lightweight adapters to transfer knowledge from the image to the video domain. Similarly, VPT (Ju et al., 2022) and Vita-CLIP (Wasim et al., 2023) leverage learnable prompts to improve the recognition performance.

Although these two types of methods have shown promise in the closed-set setting, their performance is not satisfactory in open-vocabulary settings, as shown in Fig. 1. Recently, Open-

VCLIP (Weng et al., 2023) targets the open-vocabulary setting, yet its reliance on weight interpolation constrains its applicability to networks having exactly the same weight dimensions. In contrast, our proposed framework distills knowledge from CLIP by enforcing feature consistency, without constraining the network architectures, thereby being compatible with diverse networks.

## 2.2 FEATURE-BASED KNOWLEDGE DISTILLATION

Knowledge distillation is a technique that involves transferring knowledge from a teacher model to a student model. The mainstream distillation methods can be broadly categorized as logits-based (Hinton et al., 2015), similarity-based (Tung & Mori, 2019), and feature-based (Romero et al., 2014). Feature-based distillation has been shown to provide clearer optimization targets and outperforms the other two approaches (Chen et al., 2022b).

Recently, the feature-based distillation method has been applied in CLIP-based models, *e.g.*, CLIPPING (Pei et al., 2023) and VLKD (Dai et al., 2022). CLIPPING focuses on model compression, which aims to fully transfer the knowledge from Clip4clip (Luo et al., 2022) (a large teacher model) to MobileViT-v2 (Mehta & Rastegari, 2022) (a small student model). The method is tailored for the aforementioned architectures and can not generalize to other architectures. VLKD concentrates on aligning the features of the language model to the CLIP model and subsequently integrating them to be a multi-modal generator. All of these methods aim to distill the same knowledge from one model to another. Differently, in our case, we have two objectives: maintaining the generalization capability of a pretrained CLIP and effectively adapting from image to video tasks.

To achieve our goals, we propose a novel approach, called residual feature distillation, which allows flexibly adapting features from images to videos, while not sacrificing the generalization capability of the pretrained CLIP.

## 3 METHOD

The overall pipeline of FROSTER consists of two key components, namely, model finetuning to bridge the gap between image and video tasks, and knowledge distillation to maintain the generalizability of the pretrained CLIP. In the following, we first introduce the preliminary in Sec. 3.1, followed by our method in Sec. 3.2.

### 3.1 PRELIMINARY

#### 3.1.1 ACTION RECOGNITION WITH CLIP

Owing to the strong representation stemming from pretraining on massive image-text pairs, CLIP has become a foundation model, which has been increasingly applied to various downstream tasks. Utilizing CLIP for video recognition is straightforward: one can simply use the CLIP model to process each frame individually and then average their outputs for prediction. Consider the CLIP model with ViT (Dosovitskiy et al., 2020) architecture. Given a video $x_i \in \mathbb{R}^{T \times 3 \times H \times W}$ with $T$ frames and each frame is with the spatial dimension of $H \times W$, we can directly feed it into the visual encoder $f_v$ of CLIP, which can be written as:

$$z_f^{v,i} = f_v(x_i), \tag{1}$$

where the visual feature $z_f^{v,i} \in \mathbb{R}^{T \times C}$ and $C$ denotes the feature dimension of the [CLS] token in ViT. The feature of [CLS] token embeds representation of the whole frame. To obtain the video-level representation, we employ a simple average pooling on all $z_f^{v,i}$ from different frames to obtain the global embedding vector $\bar{z}_f^{v,i} \in \mathbb{R}^C$. Regarding text processing, a common practice is to embed the action categories in pre-defined templates (*e.g.*, "*A video of []*") as the input. Consider an embedded action $t_j$, we can obtain the text features $z_f^{t,j}$ with the text encoder $f_t$ as:

$$z_f^{t,j} = f_t(t_j), \tag{2}$$

where $z_f^{t,j} \in \mathbb{R}^C$. During training, the learning objective is to maximize the image-text representation similarity between $\bar{z}_f^{v,i}$ and $z_f^{t,j}$ corresponding to the same class. However, the text data may be scarce when fine-tuning the text encoder if we just use the template-embedded action names as text inputs. To alleviate this issue, following Xiang et al. (2023); Momeni et al. (2023); Zhao et al. (2023), we adopt GPT3.5 to enrich the action names with more auxiliary descriptions, which can be viewed as text data augmentation during training. The implementation details are given in Sec. 4.

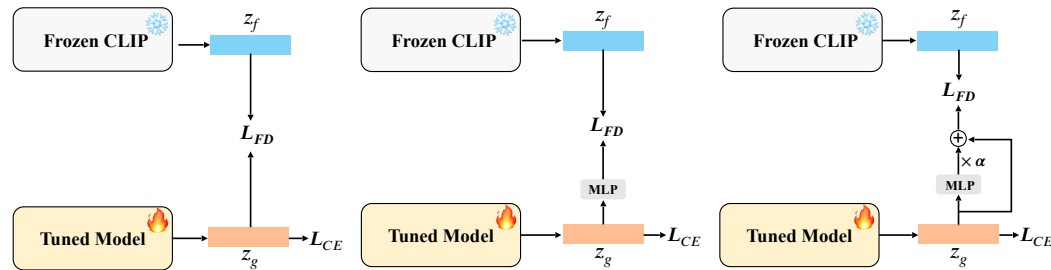

(a) Feature Distillation w/o projector.  (b) Feature Distillation w/ projector.  (c) Residual Feature Distillation.

Figure 3: Illustration of feature distillation approaches. $L_{CE}$ and $L_{FD}$ denote cross-entropy loss and feature distillation loss, respectively. (a) Directly minimizing the feature distance between the student and teacher model (Chen et al., 2022b). (b) Using a two-layer MLP to map the feature from the student space to the teacher space (Deng & Zhang, 2021; Yang et al., 2020). (c) The proposed residual feature distillation, employing a modified residual network on the student model to achieve a balance in feature learning. Our method aims to simultaneously optimize video-specific knowledge and generalizability, enhancing the overall performance of the model.

### 3.1.2 FEATURE-BASED DISTILLATION

The feature-based distillation (Romero et al., 2014; Deng & Zhang, 2021; Yang et al., 2020; Chen et al., 2022b) is achieved by enforcing the feature consistency between the teacher and student models, with loss functions such as L2 loss:

$$L_{FD} = \frac{1}{N} \sum_{i}^{N} \left\| f_v^{(t)}(x_i) - f_v^{(s)}(x_i) \right\|_2, \tag{3}$$

where $f_v^{(t)}$, $f_v^{(s)}$, and $N$ denote the teacher model, student model, and batch size respectively.

In this paper, the objective of feature distillation is to maintain generalizability in the fine-tuned model, which is crucial for open-vocabulary recognition.

### 3.2 FROSTER

**Video-specific fine-tuning.** For model fine-tuning, following Eq. (1) and Eq. (2), by feeding the video-text inputs into the tuned model $g$, we can obtain the tuned visual features $z_g^{v,i} \in \mathbb{R}^C$ and textual features $z_g^{t,j} \in \mathbb{R}^C$ as:

$$\begin{aligned} z_g^{v,i} &= g_v(x_i), \\ z_g^{t,j} &= g_t(t_j), \end{aligned} \tag{4}$$

where $g_v$ and $g_t$ denote the visual and textual encoders of the tuned model, respectively.

By calculating the feature similarities between each video and texts of all categories, we can predict the category $y_i$ for each video. Given the ground truth $\hat{y}_i$, we use a cross-entropy loss for video-specific learning:

$$L_{CE} = -\frac{1}{N} \sum_{i}^{N} \sum_{j}^{K} \hat{y}_{i,j} \log y_{i,j}, \tag{5}$$

where $K$ denotes the total number of classes.

**Residual feature distillation.** Taking the frozen CLIP as the teacher, two common ways to realize feature-based distillation are illustrated Fig. 3 (a) (Chen et al., 2022b) and Fig. 3 (b) (Deng & Zhang, 2021; Yang et al., 2020). Here, for simplicity, we use $z_g$ and $z_f$ to denote $z_g^v$ ($z_g^t$) and $z_f^v$ ($z_f^t$). As shown in Fig. 3 (a), since the output feature dimensions of the tuned model and the frozen CLIP stay the same, we can directly conduct feature distillation between them without feature projection. However, this supervision expects the tuned features $z_g$ to remain the same as the pretrained ones $z_f$, which prohibits $z_g$ from learning video-specific knowledge. Another possible way (as in Fig. 3 (b)) is to apply a projector to map $z_g$ from the student space to the teacher space. This can relax the constraints on $z_g$ for better fitting the video data. However, under such a condition, this distillation loss would be too loose for $z_g$ to keep close with $z_f$, thereby limiting its generalizability. Therefore, we need to find a trade-off between the above two methods, considering both learning objectives.

Inspired by the residual design of ResNet (He et al., 2016) for reducing optimization difficulty in deep networks, we propose a modified residual network for balancing the two learning objectives when conducting distillation. The intuition behind the design is to allow the tuned features to effectively receive supervision from generalized ones while also being video-specific. As in Fig. 3 (c), we apply a modified residual network $h$ on $z_g$ to transform its representation with a two-layer MLP projector and an identity mapping:

$$\hat{z}_g = h(z_g) = z_g + \alpha \times W_2(\sigma(W_1(z_g))), \tag{6}$$

where $W_1 \in \mathbb{R}^{C \times C}$, $W_2 \in \mathbb{R}^{C \times C}$, $\sigma$ denotes GELU (Hendrycks & Gimpel, 2016) function, and $\alpha$ is a balancing coefficient. In this way, our design enjoys three major benefits:

(1) Since there exists an identity mapping in the transformation Eq. (6), the generalizable target $z_f$ can directly guide the generalizable learning of $z_g$, which is similar to Fig. 3 (a). But differently, given the projected term $\alpha \times W_2(\sigma(W_1(z_g)))$, we do not enforce $z_g$ to be the same as $z_f$, which makes it flexible for $z_g$ to fit the video data.

(2) $\alpha$ is an important factor in balancing the learning in two objectives. If we set it as a small number, the learned embedding space for $z_g$ is largely constrained by the teacher model, otherwise $z_g$ may overfit the video data and impair generalizability. In experiments, we find that setting $\alpha$ as a relatively small number (*e.g.*, 0.1) leads to better performance than a large one. This phenomenon suggests that the pretrained CLIP already possesses strong representation, thus we only need to slightly adjust it to transfer from images to videos.

(3) Inspired by the initialization strategy in Hu et al. (2021), to make sure $\hat{z}_g$ is learned starting from the pretrained status, we initialize the parameters of the second fully connected layer $W_2$ as zeros. Therefore, at the beginning of fine-tuning, $\hat{z}_g$ only contains $z_g$ and gradually gets updated. This notably improves the training stability.

**Loss function.** After obtaining the transformed vectors $\hat{z}_g^v$ and $\hat{z}_g^t$, we can use the frozen CLIP as the teacher to distill knowledge. The distillation loss for vision and text features can be written as:

$$\begin{aligned}
L_{FD}^v &= \sum_i^N \|f_v(x_i) - h_v(g_v(x_i))\|_2, \\
L_{FD}^t &= \sum_j^K \|f_t(t_j) - h_t(g_t(t_j))\|_2, \\
L_{FD} &= L_{FD}^v + L_{FD}^t.
\end{aligned} \tag{7}$$

The overall learning objective is then the combination of classification and distillation losses:

$$L = L_{CE} + \beta L_{FD}, \tag{8}$$

where $\beta$ is a balancing coefficient.

## 4 EXPERIMENT

**Experimental setting.** Following the common practice in the literature, we adopt two experimental settings: *base-to-novel* and *cross-dataset* evaluation. *Base-to-novel*: Under this setting, for each dataset, we divide the class vocabulary into two non-overlapping sets, *i.e.*, the base set $Y_B$ and novel set $Y_N$, where $Y_B \cap Y_N = \emptyset$. The models are trained on samples from $Y_B$, and evaluated on testing samples from $Y_B \cup Y_N$. *Cross-dataset*: Under this setting, we train the models on a source dataset with the class vocabulary set as $Y_S$ and evaluate them on a target dataset with another vocabulary set as $Y_T$, where $|Y_S \cup Y_T| \geq |Y_S \cap Y_T|$. We evaluate our method using the common UCF-101 dataset (Soomro et al., 2012), HMDB-51 dataset (Kuehne et al., 2011), Kinetics-400 (K-400) dataset (Carreira & Zisserman, 2017), Kinetics-600 (K-600) dataset (Carreira et al., 2018), and Something-to-Something V2 (SSv2) dataset (Goyal et al., 2017). K-400 and K-600 are large-scale action recognition datasets with 400 and 600 action classes, respectively. UCF-101 consists of 13,320 video clips from 101 action classes, and HMDB-51 includes 6,849 videos from 51 action classes. SSv2 contains 174 fine-grained action classes. We follow the literature (*e.g.*, Rasheed et al. (2023); Ni et al. (2022); Weng et al. (2023)) to conduct experiments under the two settings and report the average top-1 accuracy.

Table 1: Performance comparison (Top1-Acc (%)) with the CLIP-based methods using ViT-B/16 under the base-to-novel setting. "HM" denotes the harmonic mean of the accuracy from the base and novel sets. The results of most other papers are taken from ViFi-CLIP. † denotes the results with our implementation. The best results are **bolded**, and the second-best results are underlined.

| Method | K-400 Base | Novel | HM | HMDB-51 Base | Novel | HM | UCF-101 Base | Novel | HM | SSv2 Base | Novel | HM |
|---|---|---|---|---|---|---|---|---|---|---|---|---|
| Frozen CLIP (Radford et al., 2021) | 62.3 | 53.4 | 57.5 | 53.3 | 46.8 | 49.8 | 78.5 | 63.6 | 70.3 | 4.9 | 5.3 | 5.1 |
| ActionCLIP (Wang et al., 2021) | 61.0 | 46.2 | 52.6 | 69.1 | 37.3 | 48.5 | 90.1 | 58.1 | 70.7 | 13.3 | 10.1 | 11.5 |
| XCLIP (Ni et al., 2022) | 74.1 | 56.4 | 64.0 | 69.4 | 45.5 | 55.0 | 89.9 | 58.9 | 71.2 | 8.5 | 6.6 | 7.4 |
| VPT (Ju et al., 2022) | 69.7 | 37.6 | 48.8 | 46.2 | 16.0 | 23.8 | 90.5 | 40.4 | 55.8 | 8.3 | 5.3 | 6.4 |
| AIM † (Yang et al., 2023) | 74.6 | 62.5 | 68.0 | 64.0 | 51.6 | 57.1 | 89.8 | 76.4 | 82.6 | 8.5 | 7.9 | 8.2 |
| ST-Adapter † (Yang et al., 2023) | 73.6 | 62.0 | 67.3 | 65.3 | 48.9 | 55.9 | 85.5 | 76.8 | 80.9 | 9.3 | 8.4 | 8.8 |
| ViFi-CLIP (Rasheed et al., 2023) | 76.4 | 61.1 | 67.9 | 73.8 | 53.3 | 61.9 | 92.9 | 67.7 | 78.3 | 16.2 | 12.1 | 13.9 |
| Open-VCLIP † (Weng et al., 2023) | 76.5 | 62.6 | 68.9 | 70.3 | 50.4 | 58.7 | 94.8 | 77.5 | 85.3 | 16.0 | 11.0 | 13.0 |
| **FROSTER** | **77.8** | **64.3** | **70.4** | **74.1** | **58.0** | **65.1** | **95.3** | **80.0** | **87.0** | **18.3** | **12.2** | **14.6** |

**Architectures.** We use CLIP with ViT-B/16 vision encoder for all experiments. As for the tuned model, we adopt VCLIP (Weng et al., 2023), fully fine-tuned CLIP, Action CLIP (Wang et al., 2021), Adaptformer (Chen et al., 2022a), AIM (Yang et al., 2023), and ST-Adapter (Pan et al., 2022) to verify the effectiveness of our method. Unless stated otherwise, we use VCLIP for our experiments.

**Text augmentation.** To enrich the category names with more details for training, we adopt GPT3.5 to generate helpful descriptions using the instruction: "*Please describe this action in the video []*". We find that such a simple instruction is effective in providing extra text for training. For instance, the action "*brushing teeth*" is augmented to "*The video teaches the process of brushing teeth, which involves using a toothbrush and toothpaste to clean and maintain oral hygiene*".

More experimental details can be found in Appendix A.1.

## 4.1 COMPARISON WITH STATE-OF-THE-ART METHODS

**Base-to-novel.** In Tab. 1, we compare our method with the previous methods under the base-to-novel setting. From the results, three notable findings are summarized: (1) Compared with some adapted models (Action CLIP, X-CLIP and VPT), the frozen CLIP is still competitive, especially on the novel sets of K-400, HMDB-51, and UCF-101, indicating its strong generalizability. (2) FROSTER achieves the best performance over all datasets for both the base and novel sets, validating its capacity to be video-specific and generalizable. (3) Compared to the baseline (VCLIP), our method can achieve consistent achievements on all base and novel categories, which further verifies its effectiveness. (4) On K-400, HMDB-51, and UCF-101, FROSTER achieves larger improvements on the novel sets over the base sets. This demonstrates its great potential to be applied in open-world applications. Besides, we observe that the performance gains achieved on the novel set of SSv2 are relatively modest compared to other datasets. Given the fine-grained nature of SSv2, it requires a stronger temporal awareness of the model to perform well on this dataset. As no cross-frame temporal information was used during the CLIP pretraining, it is still challenging for CLIP to generalize well on the fine-grained actions. This could potentially also explain why the performance of all other methods is significantly lower on SSv2 compared to other datasets.

**Cross-dataset.** In Tab. 2, we compare our method with the previous methods under the cross-dataset setting. We notice that, on the most generalizability demanding dataset, *i.e.*, K-600, the frozen CLIP outperforms most of the other methods, further demonstrating its superior generalizability over the adapted video CLIP models. FROSTER achieves the best performance over all datasets compared with the fully-finetuning methods (ActionCLIP and ViFi-CLIP), adapter-based methods (X-CLIP, AIM, and ST-Adapter), prompting method (VPT), and weight interpolation method (Open-VCLIP), demonstrating its strong generalizabilty.

## 4.2 DIAGNOSTIC STUDY

**Effectiveness with different networks.** In Tab. 3, we apply FROSTER with two types of CLIP-based video models, *i.e.*, fully-tuned and adapter-based. We notice that: (1) For all the networks, FROSTER can effectively improve performance, highlighting its broad applicability. Empirically, we find that FROSTER achieves larger improvements on K-600, which is a more generalizability-

Table 2: Performance comparison (Top1-Acc (%)) with the previous approaches under the cross-dataset setting. All methods are based on CLIP ViT-B/16, except for ER-ZASR (TSM (Lin et al., 2019) pre-trained on ImageNet-1k) and Text4Vis (ViT-L/14). UCF* and HMDB* indicate evaluating the full validation set, while UCF and HMDB denote evaluating across the three validation splits. The results of most other papers are taken from Open-VCLIP and ViFi-CLIP. † denotes the results are produced with our implementation.

| Method | UCF* | UCF | HMDB* | HMDB | K-600 |
|---|---|---|---|---|---|
| ER-ZASR (Chen & Huang, 2021) | - | 51.8±2.9 | - | 35.3±4.6 | 42.1±1.4 |
| Frozen CLIP † (Radford et al., 2021) | 74.2 | 73.8±0.6 | 46.3 | 47.9±0.5 | 68.1±1.1 |
| ActionCLIP † (Wang et al., 2021) | 77.4 | 77.5±0.8 | 48.0 | 48.2±1.5 | 62.5±1.2 |
| X-CLIP (Ni et al., 2022) | - | 72.0±2.3 | - | 44.6±5.2 | 65.2±0.4 |
| VPT (Ju et al., 2022) | - | 69.3±4.2 | - | 44.3±2.2 | 55.8±0.7 |
| Text4Vis (Wu et al., 2023) | 79.6 | - | 49.8 | - | 68.9±1.0 |
| AIM † (Yang et al., 2023) | 79.0 | 79.4±1.0 | 49.5 | 50.3±0.8 | 66.7±0.5 |
| ST-Adapter † (Pan et al., 2022) | 77.9 | 77.6±0.7 | 50.3 | 51.1±0.6 | 60.2±1.8 |
| Vita-CLIP (Wasim et al., 2023) | - | 75.0±0.6 | - | 48.6±0.6 | 67.4±0.5 |
| ViFi-CLIP (Rasheed et al., 2023) | - | 76.8±0.7 | - | 51.3±0.6 | 71.2±1.0 |
| Open-VCLIP (Weng et al., 2023) | 83.5 | 83.4±1.2 | 53.2 | 53.9±1.2 | 73.0±0.8 |
| **FROSTER** | **85.0** | **84.8**±1.1 | **54.5** | **54.8**±1.3 | **74.8**±0.9 |

Table 3: Performance comparison (top1-Acc (%)) when using FROSTER with different networks.

| | Method | UCF* | UCF | HMDB* | HMDB | K-600 |
|---|---|---|---|---|---|---|
| Fully-Tuned | Fine-tuned CLIP | 80.0 | 80.0±0.8 | 48.3 | 49.8±1.6 | 66.1±1.1 |
| | **Fine-tuned CLIP w/FROSTER** | **83.5** | **83.3**±0.8 | **53.6** | **53.8**±1.5 | **73.6**±1.0 |
| | Action CLIP (Wang et al., 2021) | 77.4 | 77.5±0.8 | 48.0 | 48.2±1.5 | 62.5±1.2 |
| | **Action CLIP w/ FROSTER** | **84.0** | **83.7**±0.8 | **54.0** | **54.1**±0.8 | **73.8**±0.9 |
| | VCLIP (Weng et al., 2023) | 79.7 | 79.8±1.0 | 49.8 | 50.3±0.8 | 65.9±1.0 |
| | **VCLIP w/FROSTER** | **85.0** | **84.8**±1.1 | **54.5** | **54.8**±1.3 | **74.8**±0.9 |
| Adapter-Based | AdapterFormer (Chen et al., 2022a) | 80.5 | 80.3±1.0 | 50.5 | 51.0±0.8 | 67.0±0.4 |
| | **AdapterFormer w/FROSTER** | **81.0** | **80.8**±1.2 | **53.1** | **53.5**±1.3 | **74.0**±0.9 |
| | AIM (Yang et al., 2023) | 79.0 | 79.4±1.0 | 49.5 | 50.3±0.8 | 66.7±0.5 |
| | **AIM w/FROSTER** | **81.2** | **81.0**±1.3 | **53.2** | **53.5**±1.3 | **74.1**±0.8 |
| | ST-Adapter (Pan et al., 2022) | 77.9 | 77.6±0.7 | 50.3 | 51.1±0.6 | 60.2±1.8 |
| | **ST-Adapter w/FROSTER** | **79.5** | **79.3**±1.1 | **52.0** | **52.5**±1.3 | **73.1**±0.9 |

demanding dataset compared with UCF-101 and HMDB-51. (2) Combining with FROSTER, fully fine-tuned models can generally achieve better results than the adapter-based methods. This improvement might be attributed to the fitting capacity of more trainable parameters, which afford greater flexibility for attaining both video-specific and generalizable learning objectives than the adapter-based models.

**Effectiveness of residual feature distillation.** In Tab. 4, we conduct experiments to verify the effectiveness of our distillation design (see Fig. 3). It is notable that: (1) For both the vision and text encoders when applying distillation methods (see Exp. B - D and Exp. F - H), they can effectively improve recognition performance compared with not using them (see Exp. A and Exp. E). This phenomenon verifies the useful generalizable knowledge in the frozen CLIP. (2) By comparing Exp. B and C (or Exp. F and G), we find that using a projector for feature distillation achieves inferior performance than not using it, which indicates that the projector may hinder the distillation supervision, which degrades the generalizable capacity. (3) Using the proposed residual feature distillation method achieves the best performance (see Exp. D and Exp. H), verifying its superiority over the other two widely used methods for balancing video-specific and generalizable learning. (4) By comparing Exp. I and J, we can see that the simple class name enriching technique can effectively improve the action recognition performance.

**Attention visualization.** As shown in Fig. 4, we notice that frozen CLIP tends to focus on the background regions instead of the moving objects. In contrast, the fine-tuned VCLIP exhibits attention towards the key semantic elements (*e.g.*, bottle and cup). Further, our method attends to the moving parts more (*e.g.*, hands), while considering also the helpful background information (*e.g.*, sink and table).

Table 4: Effects of frozen CLIP distillation methods and the augmented text under the cross-dataset evaluation setting. "RFD" denotes Residual Feature Distillation.

| Exp. | Tuned Encoder | Distillation w/o MLP | w/ MLP | RFD | UCF* | HMDB* | K-600 |
|------|---------------|----------------------|--------|-----|------|-------|-------|
| A | Vision | ✓ | | | 79.7 | 49.8 | 65.9±1.0 |
| B | Vision | ✓ | | | 81.8 | 51.8 | 70.1±0.9 |
| C | Vision | | ✓ | | 80.5 | 49.8 | 67.9±0.9 |
| D | Vision | | | ✓ | **82.9** | **52.7** | **71.1**±1.0 |
| E | Text | ✓ | | | 77.2 | 48.5 | 69.3±0.9 |
| F | Text | ✓ | | | 78.3 | 51.7 | 70.6±0.9 |
| G | Text | | ✓ | | 78.2 | 48.7 | 69.7±1.0 |
| H | Text | | | ✓ | **78.6** | **52.3** | **71.0**±0.9 |
| I | Vision + Text | | | ✓ | 84.4 | 54.4 | 73.0±0.7 |
| J | Vision + Text (Text Augmented) | | | ✓ | **85.0** | **54.5** | **74.8**±0.9 |

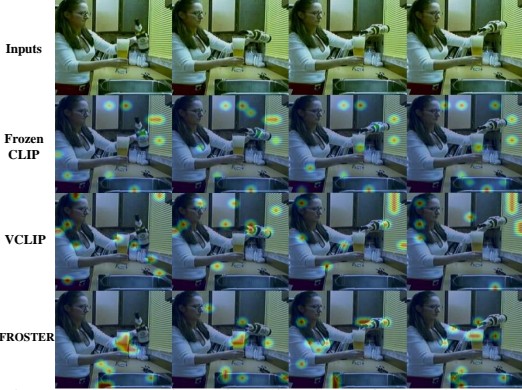

Figure 4: Attention visualization of attention correlations between `[CLS]` and image tokens of the action "pour". Our method focuses on moving objects and informative backgrounds.

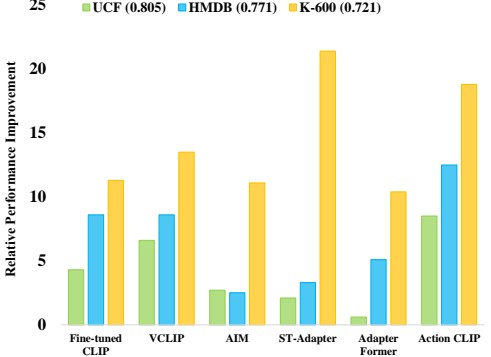

Figure 5: Relative improvements of different methods when combined with FROSTER. The number next to the legend for each dataset indicates the dataset's similarity with K-400 using Hausdorff distance (cosine similarity).

**Dataset semantic distance v.s. performance improvements.** To study the relationship between dataset semantic distances and performance improvements, we measure the semantic distances between the training and testing datasets by comparing text features of action class names between datasets. Figure 5 presents the relative improvements on different testing datasets by different models. We observe that greater improvements are attained on datasets that require more generalizability, *i.e.*, K-600. This finding serves as additional evidence of the efficacy of FROSTER in enhancing generalizable capacity.

Please refer to the appendix for additional experiments and visualizations.

## 5 CONCLUSION

In this paper, we present FROSTER, a simple yet effective framework designed to tackle the open-vocabulary video recognition task. FROSTER achieves the dual objective of being video-specific and generalizable. To accomplish this, we introduce a frozen CLIP model as a teacher model to facilitate generalization. Additionally, we propose a residual feature distillation method to balance feature learning in both objectives. FROSTER is compatible with various network architectures and has been extensively evaluated on base-to-novel and cross-dataset evaluation settings using large-scale video datasets, demonstrating its effectiveness.

## ACKNOWLEDGMENTS

This work is supported by Hong Kong Research Grant Council - Early Career Scheme (Grant No. 27208022), National Natural Science Foundation of China (Grant No. 62306251), and HKU Seed Fund for Basic Research.

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

# A  APPENDIX

## A.1  MORE EXPERIMENTAL DETAILS

**Datasets.**    We evaluate our method using the common UCF-101 (Soomro et al., 2012), HMDB-51 (Kuehne et al., 2011), Kinetics-400 (K-400) (Carreira & Zisserman, 2017), Kinetics-600 (K-600) (Carreira et al., 2018), and Something-to-Something V2 (SSv2) (Goyal et al., 2017) datasets. (1) *K-400 & K-600* are large-scale action recognition datasets, containing 400 and 600 action classes, respectively. K-400 includes 240k training and 20k validation samples, while the K-600 dataset is an extension of the K-400, which has 410k training and 29k validation samples. K-600 contains 220 non-overlapped categories compared to K-400. Each sample is a YouTube video clip representing a human action. (2) *UCF-101* consists of 13,320 video clips from 101 action classes, where 9,537 samples are used as the training set and the remaining 3,783 samples are used for testing. Officially, there are three training/testing splits. The dataset includes a variety of human activities and is widely used for benchmarking in human action recognition research. The videos vary in length but are generally short. (3) *HMDB-51* (Kuehne et al., 2011) includes 6,849 videos from 51 action classes and has more than 101 samples per class. Similar to UCF-101, there are also three official training/testing splits. The dataset comprises a diverse range of sources, including movies, public databases, and YouTube videos, offering a broad range of human actions and interactions. (4) *SSv2* contains 174 fine-grained action classes, which are mostly related to actions performed with daily objects. In total, there are 168,913 video clips for training and 24,777 video clips for testing. This dataset is unique in its focus on human-object dynamics, thereby is known as a temporal-challenging dataset.

**Evaluation metrics.**    We follow the literature (*e.g.*, Rasheed et al. (2023); Ni et al. (2022); Weng et al. (2023)) to conduct experiments under the two experimental settings. For the *base-to-novel* setting, we conduct experiments on UCF-101, HMDB-51, K-400, and K-600, and report the average top-1 accuracy. The action classes are separated into base and novel classes for each dataset. Frequent classes are used as the base classes while less frequent classes are used as novel classes. The models are trained on samples from base classes in the raw training split in each dataset and evaluated on samples from novel classes in the raw validation split. 16 video clips are sampled for each base class for training. Three different base sets are constructed for each dataset and used to train the model separately. The resulting models are evaluated on the same novel set and we report the average results using models trained on three different base sets. During testing, HMDB-51 and UCF-101 datasets have three validation splits in the raw data. However, in the base-to-novel setting being used here, only the samples from novel classes in the first split are used for evaluation. On the other hand, K-400 and SSv2 datasets have only one validation split in the raw data, and the samples from the novel classes in the entire split are used for evaluation here. For the *cross-dateset setting*, the models are trained on K-400 (Carreira & Zisserman, 2017), and evaluated on UCF-101 (Soomro et al., 2012), HMDB-51 (Kuehne et al., 2011), and K-600 (Carreira et al., 2018). For HMDB-51 and UCF-101, the methods are evaluated using their respective three validation splits in the raw data, and we report the average top-1 accuracy on these splits as well as the performance variance. For K-600, the methods are evaluated on the 220 categories that do not exist in K-400. We report the average top-1 accuracy over three randomly sampled splits used in Ni et al. (2022); Rasheed et al. (2023); Weng et al. (2023), with each split containing 160 categories.

**Training configurations.**    The initial learning rate is set to $3.33 \times 10^{-6}$ and is decayed using the cosine scheduler. For base-to-novel evaluation, we train each model for 12 epochs and set the first 2 epochs for warming up. Differently, for cross-dataset evaluation, since we have larger training data, we train the models for 22 epochs with the first 2 epochs as a warm-up. The hyper-parameters $\alpha$ and $\beta$ are set as 0.1 and 2. During training, each video is uniformly sampled with 8 frames. During testing, we sample 3 video clips (8 frames per clip) with 1 crop ("$3 \times 1$" views) of each video and ensemble the outputs with an average summation. Following Open-VCLIP (Weng et al., 2023), when testing, we average the models learned in different epochs to improve generalizability for the cross-dataset setting. We use $8 \times$ A100 GPUs to conduct all the experiments.

## A.2 DIAGNOSTIC EXPERIMENTS

**Impacts of the zero initialization.** As shown in Tab. 5, we notice that zero initialization can improve the performance since it enables a starting point just like the pretrained CLIP which gets updated smoothly.

**Impacts of the weight $\alpha$.** As shown in Tab. 6, we conduct experiments by setting $\alpha$ to 1.0, 0.5, 0.1, and 0.05, respectively. We find that smaller $\alpha$ generally obtains better results, which indicates that the video-specific learning should be well-constrained. For the performance trade-off on different datasets, we set $\alpha$ as 0.1.

Table 5: Impacts of the zero initialization of $W_2$.

| Method | UCF* | HMDB* | K-600 |
|---|---|---|---|
| Ours *w/o* Zero Init. | 84.7 | 54.1 | 74.4±1.0 |
| Ours *w/* Zero Init. | **85.0** | **54.5** | **74.8**±0.9 |

Table 6: Impacts the value of $\alpha$.

| $\alpha$ | UCF* | HMDB* | K-600 |
|---|---|---|---|
| 1.0 | 84.7 | 53.1 | 73.2±1.0 |
| 0.5 | 84.8 | 53.9 | 74.0 ±0.9 |
| **0.1** | 85.0 | 54.5 | **74.8**±0.9 |
| 0.05 | **85.1** | **54.8** | 74.3±0.9 |

Table 7: Impacts of the balancing coefficient $\beta$ in the loss function.

| $\beta$ | UCF* | HMDB* | K-600 |
|---|---|---|---|
| 1 | 84.7 | 54.3 | 74.0±1.0 |
| 2 | **85.0** | **54.8** | 74.8±0.9 |
| 3 | 85.0 | 54.1 | **75.0**±**1.0** |

**Impacts of the coefficients $\beta$.** In Tab. 7, we experiment with different values for $\beta$ by setting it to 1, 2, and 3, respectively. We can see that the variance among different values is not large and choose $\beta = 2$ to be our default choice.

## A.3 VISUALIZATION RESULTS

In Fig. 6-Fig. 9, we present more qualitative comparison. Overall, our model attends to informative regions related to the action for more reliable recognition. For example, in Fig. 6, our method effectively captures the movements of the woman's mouth to extract the motion features relevant to "chew"; in Fig. 7, our model attends to the human legs when recognizing "push".

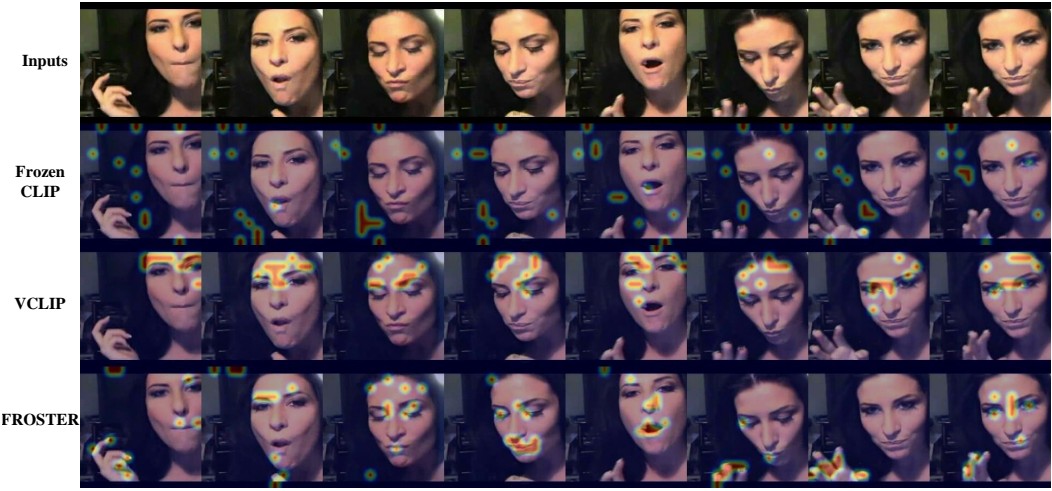

Figure 6: Attention maps for "chew". (1) The frozen CLIP attends to the background. (2) The tuned VCLIP attends to the face and forehead. (3) FROSTER attends to the mouth, which is more relevant to the action.

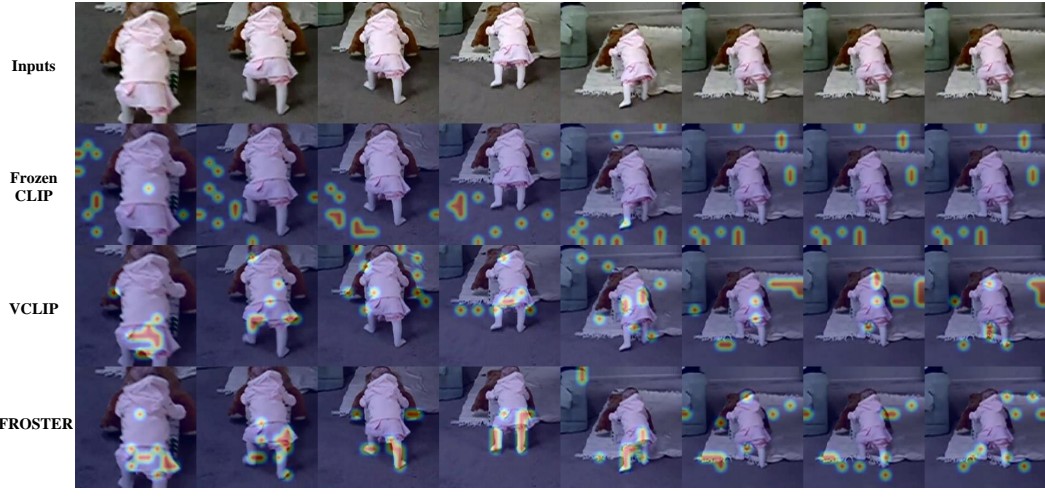

Figure 7: Attention maps for "push". (1) The frozen CLIP attends to the surrounding stuff around the baby. (2) The tuned VCLIP attends to the clothing of the baby. (3) FROSTER attends to the legs of the baby, which are more relevant to the action.

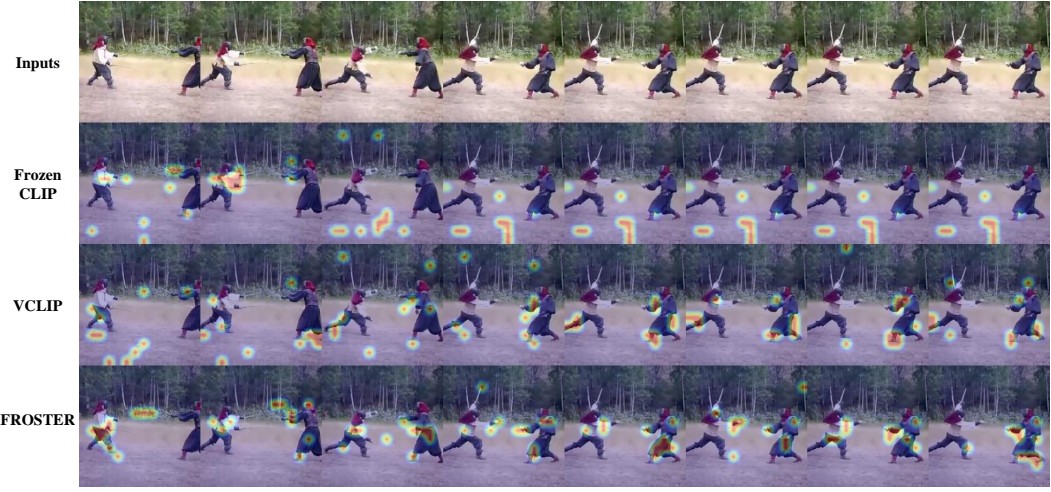

**Fencing**

Figure 8: Visualization of attention maps for action "fencing". (2) The frozen CLIP attends to the background, *e.g.*, the ground. (2) The tuned VCLIP learns to focus on the moving body parts. (3) FROSTER attends to the arms and legs, which are more relevant to the action.

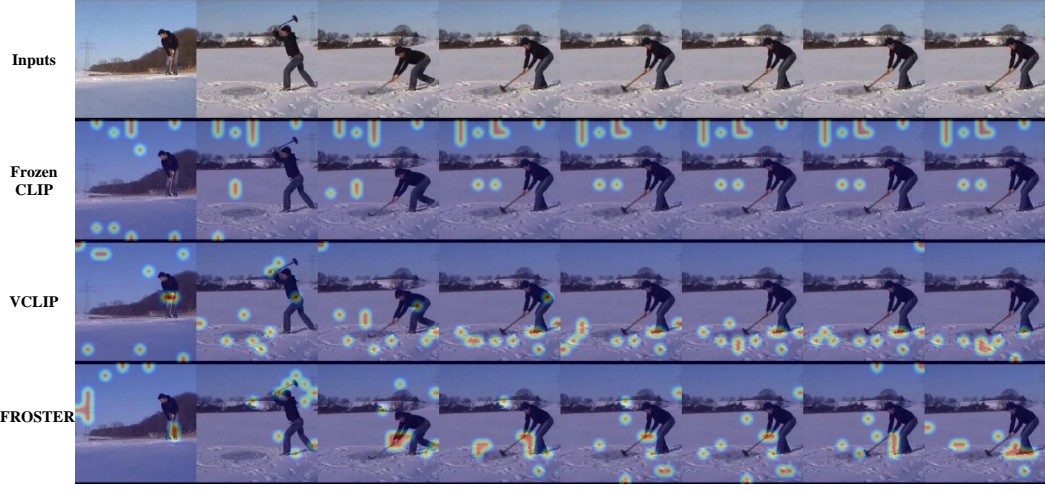

**Hit**

Figure 9: Attention maps for "hit". (1) The frozen CLIP attends to the ground and sky. (2) The tuned VCLIP attends to the person's pelvic region and the ground. (3) FROSTER attends to the person's hands and the hammer, which are more relevant to the action.

