# FROSTER: FROZEN CLIP IS A STRONG TEACHER FOR OPEN-VOCABULARY ACTION RECOGNITION (*Supplementary Material*)

## 1 FEATURE VISUALIZATION

We include the t-SNE visualizations of the learned embeddings on both base and novel datasets in Fig. 1. Interestingly, we observe that despite the superior performance of our method compared to the VCLIP model (or superior performance of VCLIP over frozen CLIP), there does not appear to be much visual discrepancy in the t-SNE projection of different methods. The possible reason is that the fine-grained spatial-temporal information that is essential for discriminating actions may not be well preserved in the low-dimensional space of t-SNE. However, from the visualizations of spatial-temporal attention maps (refer to Fig. 4 in our paper and Fig.6, Fig. 7, Fig. 8, and Fig. 9 in the Appendix), our method can better focus on parts related to informative patterns and actions compared to frozen CLIP and VCLIP.

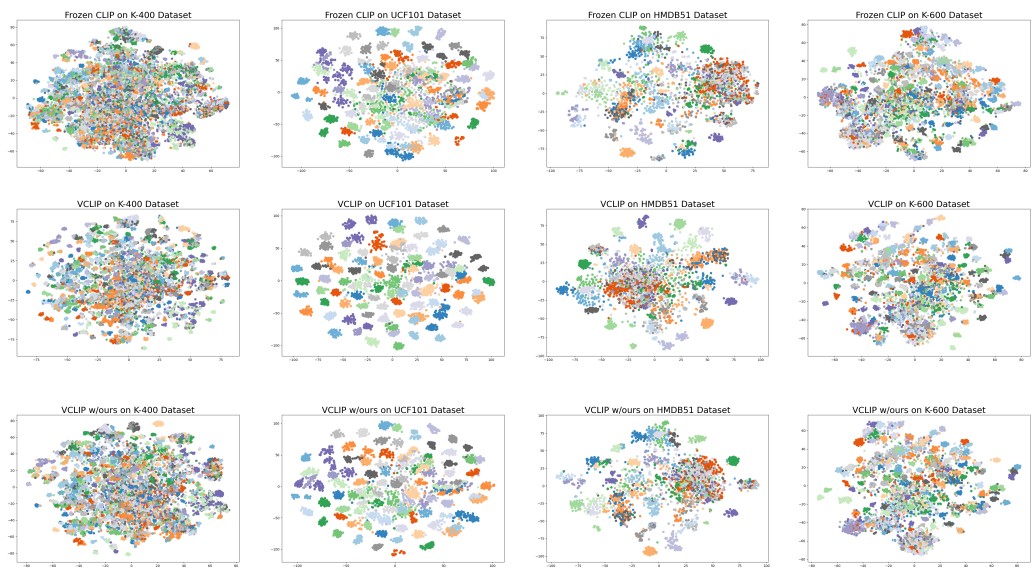

Figure 1: t-SNE Visualizations of Learned Embeddings. We visualize the learned embedding of the videos on the base dataset (K-400) and the novel datasets (UCF101, HMDB51, and K-600).