# OpenReview forum: "FROSTER: Frozen CLIP is A Strong Teacher for Open-Vocabulary Action Recognition"
_ICLR.cc/2024/Conference — ICLR 2024 poster_

### Official Review · Reviewer_RLyA · 2023-10-31

**Soundness:** 3 good
**Presentation:** 3 good
**Contribution:** 3 good
**Rating:** 6
**Confidence:** 4

**Summary:**

In this paper, the authors present FROSTER, a clip-centric approach tailored for open vocabulary action recognition. They propose a simple but effective residual feature distillation and the proposed method achieves state-of-the-art results on five datasets in both base-to-novel and cross-dataset scenarios.

**Strengths:**

1. The proposed residual feature distillation is simple but effective.

2. The proposed method achieves promising performance on five datasets under both cross-dataset and base-to-novel settings.

**Weaknesses:**

My major concerns are on the experimental part.

1. The authors touch upon "Text rephrasing" in Sec. 4.2, yet there's an absence of any ablation study related to it.

2. In the caption of Table 1, the authors mentioned that 'The results of most of the other papers are taken from Open-VCLIP and
ViFi-CLIP.', where are the results for Open-VCLIP? From Table 2 and Table 3, if my understanding is correct, the reported results use the Open-VCLIP as the backbone. Hence, showcasing Open-VCLIP's results in Table 1 becomes critical to underscore the effectiveness of the suggested method.

3. It would be beneficial to incorporate columns for "encoder" and "pretrained dataset" within the table for clearer comprehension.

**Questions:**

Please refer to 'weakness'.

**Details Of Ethics Concerns:**

NIL

---

> ### Author Response · Authors · 2023-11-20
> **Response to Reviewer RLyA**
>
> Thanks for your precious time reviewing this paper and writing meticulous and constructive feedback. We are delighted that this paper is found to be simple but effective, and the experimental results are promising.
>
> >**Q1:** The authors touch upon "Text rephrasing" in Sec. 4.2, yet there's an absence of any ablation study related to it.
>
> **R:** The impact of text rephrasing is shown in the last two rows of Tab. 4 of the paper, which is presented below. It can be found that rephrasing the class names into descriptions can improve performance since they provide more detailed information about the actions. The corresponding experimental analysis is given in Sec. 4.3 of the paper.
>
> **Table 1: Impacts of text rephrasing under the cross-dataset evaluation setting.**
> | Method | UCF101 | HMDB51 | K-600 |
> | :------: | :------: | :------: | :------: |
> | Ours w/o text rephrasing | 84.4 | 54.4 | 73.0 $\pm$ 0.7 |
> | Ours | **85.0** | **54.5** | **74.8** $\pm$ 0.9 |
>
> >**Q2:** In the caption of Table 1, the authors mentioned that 'The results of most of the other papers are taken from Open-VCLIP and ViFi-CLIP.', where are the results for Open-VCLIP? From Table 2 and Table 3, if my understanding is correct, the reported results use the Open-VCLIP as the backbone. Hence, showcasing Open-VCLIP's results in Table 1 becomes critical to underscore the effectiveness of the suggested method.
>
> **R:** Please refer to Tab. 1 of our paper. Thanks for the suggestion and we have included the suggested comparison. Below, we present the results of VCLIP. The results demonstrate that our method can improve the performances for both the base and novel categories for all datasets.
>
> **Table 2: Performance (top1-acc (\%)) under the base-to-novel evaluation setting. B, N, and HM denote the base set, novel set, and harmonic mean, respectively.**
> | Method | K400(B) | K400(N) | K400(HM) | UCF(B) | UCF(N) | UCF(HM) | HMDB(B) | HMDB(N) | HMDB(HM) | SSv2(B) | SSv2(N) | SSv2(HM)$ |
> | :------: | :------: | :------: | :------: | :------: | :------: | :------: | :------: | :------: | :------: | :------: | :------: | :------: |
> | VCLIP | 76.5 | 62.6 | 68.9 | 70.3 | 50.4 | 58.7 | 94.8 | 77.5 | 85.3 | 16.0 | 11.0 | 13.0 |
> | **VCLIP w/ours** | **77.8** | **64.3** | **70.4** | **74.1** | **58.0** | **65.1** | **95.3** | **80.0** | **87.0** | **18.3** | **12.2** | **14.6** |
>
> >**Q3:** It would be beneficial to incorporate columns for "encoder" and "pretrained dataset" within the table for clearer comprehension.
>
> **R:** We have followed this suggestion to indicate this information in the table.
>
> Thanks again for your valuable comments and looking forward to your reply.
>
> [1] Chen, Shizhe, and Dong Huang. "Elaborative rehearsal for zero-shot action recognition." Proceedings of the IEEE/CVF International Conference on Computer Vision. 2021.
>
> [2] Lin, Ji, Chuang Gan, and Song Han. "Tsm: Temporal shift module for efficient video understanding." Proceedings of the IEEE/CVF International Conference on Computer Vision. 2019.

---

> > ### Comment · Reviewer_RLyA · 2023-11-22
> > **Discussion**
> >
> > The authors' response effectively tackled my concerns. I will maintain my rating as 'weak accept.'

---

> > > ### Author Response · Authors · 2023-11-22
> > > **Reply to the feedback from Reviewer RLyA**
> > >
> > > Thanks for your recognition and kind words about our work. Your appreciation inspires us a lot.

---

### Official Review · Reviewer_FMt3 · 2023-10-31

**Soundness:** 3 good
**Presentation:** 3 good
**Contribution:** 2 fair
**Rating:** 6
**Confidence:** 3

**Summary:**

This paper presents a CLIP-based framework for open-vocabulary action recognition, which distills the CLIP features by treating it as a strong teacher.

**Strengths:**

1. The paper is well-written and easy to follow.

2. The proposed Residual Feature Distillation is interesting and inspiring and outperforms the previous methods by a large margin (As shown in Table 1 and 2).

**Weaknesses:**

1. It would be better to report the finetuned results on all datasets (including K-400) for a better comparison.

2. The performance gains compared to baseline methods (in Table 1 and 2) are indeed surprising, and it would be better to visualize the learned embeddings on both base and novel datasets to see the effects of RFD on feature learning.

**Questions:**

Please refer to the Weakness. In addition, I still have a few questions:

1. The results in Table 1 are a little bit confusing, as the "Top1 accuracy" is only 77.8 on base categories. It would be better to further clarify the training and evaluation protocol, and the number of clips/crops used.

2. As the paper employs rephrased action descriptions, instead of action names, it would be interesting to show the video-text retrieval results.

---

> ### Author Response · Authors · 2023-11-20
> **Response to Reviewer FMt3 (1/2)**
>
> We are grateful for your thoughtful and thorough review. Your insightful comments and constructive suggestions are of great help to us. We are delighted that our paper is found to be well-written and easy to follow, the proposed method is interesting, and the performance improvement is significant.
> Please refer to the detailed response below regarding the raised several weaknesses.
>
> >**Q1:** It would be better to report the finetuned results on all datasets (including K-400) for a better comparison.
>
> **R:** Thanks for the kind suggestion and we have included the additional results in Tab. 5 of Appendix A.3. Since this paper focuses on the open-vocabulary recognition setting, we did not include the closed-set recognition results (fine-tuning and evaluation are conducted on the same dataset) in the initial submission. In Tab. 1 below, we include the closed-set evaluation results on K-400. We can see that our method achieves comparable performance to the state-of-the-art closed-set models (AIM and ViFi-CLIP). At the same time, our method consistently outperforms them by a large margin under the cross-dataset evaluation, i.e., UCF101, HMDB51, and K-600 datasets. The results indicate that the state-of-the-art closed-set methods fit the seen categories well but cannot be satisfactorily generalized to unseen ones. To evaluate the overall performance under the closed-set and open-vocabulary settings, we report the harmonic mean of them in Tab. 2 below. From Tab. 2, we find that our method achieves the best trade-off on both the closed-set and open-vocabulary settings.
>
> By the way, inspired by this suggestion, developing a more comprehensive protocol to evaluate overall performance on both closed-set and cross-dataset scenarios is a promising direction for our future work.
>
> **Table 1: Performance (top1-acc (\%)) of different methods under the closed-set (K-400) and cross-dataset (UCF,HMDB, and K-600) evaluation settings. Except for ER-ZASR (TSM), all the methods are based on the CLIP-ViT-B/16 model.**
>
> | Method | K-400 | UCF101* | UCF101 | HMDB51* | HMDB51 | K-600 |
> | :------: | :------: | :------: | :------: | :------: | :------: | :------: |
> ER-ZASR | - | - | 51.8 $\pm$ 2.9 | - | 35.3 $\pm$ 4.6 | 42.1 $\pm$ 1.4 |
> Frozen CLIP | 57.2 | 74.2 | 73.8 $\pm$ 0.6 | 46.3 | 47.9 $\pm$ 0.5 | 68.1 $\pm$ 1.1 |
> ActionCLIP | 82.6 | 77.4 | 77.5 $\pm$ 0.8 | 48.0 | 48.2 $\pm$ 1.5 | 62.5 $\pm$ 1.2 |
> X-CLIP | 83.8 | - | 72.0 $\pm$ 2.3 | - | 44.6 $\pm$ 5.2 | 65.2 $\pm$ 0.4 |
> VPT | 76.9 | - | 69.3 $\pm$ 4.2 | - | 44.3 $\pm$ 2.2 | 55.8 $\pm$ 0.7 |
> Text4Vis | 82.9 | 76.4 | 76.4 $\pm$ 0.6 | 44.5 | 46.1 $\pm$ 0.2 | 60.1 $\pm$ 0.5 |
> AIM | **83.9** | 79.0 | 79.4 $\pm$ 1.0 | 49.5 | 50.3 $\pm$ 0.8 | 66.7 $\pm$ 0.5 |
> ST-Adapter | 82.0 | 77.9 | 77.6 $\pm$ 0.7 | 50.3 | 51.1 $\pm$ 0.6 | 60.2 $\pm$ 1.8 |
> Vita-CLIP | 81.8 | - | 75.0 $\pm$ 0.6 | - | 48.6 $\pm$ 0.6 | 67.4 $\pm$ 0.5 |
> ViFi-CLIP | **83.9** | - | 76.8 $\pm$ 0.7 | - | 51.3 $\pm$ 0.6 | 71.2 $\pm$ 1.0 |
> Open-VCLIP | 81.8 | 83.5 | 83.4 $\pm$ 1.2 | 53.2 | 53.9 $\pm$ 1.2 | 73.0 $\pm$ 0.8 |
> **Ours** | 82.0 | **85.0** | **84.8** $\pm$ 1.1 | **54.5** | **54.8** $\pm$ 1.3 | **74.8** $\pm$ 0.9 |
>
> **Table 2: Harmonic mean (top1-acc (\%)) of different methods under the closed-set (K-400) and cross-dataset (UCF,HMDB, and K-600) evaluation settings.**
>
> | Frozen CLIP | Action CLIP | X-CLIP | VPT | Text4Vis | AIM | ST-Adapter | Vita-CLIP | ViFi-CLIP | Open-VCLIP | **Ours** |
> | :------: | :------: | :------: | :------: | :------: | :------: | :------: | :------: | :------: | :------: | :------: |
> | 60.1 | 64.8 | 62.9 | 58.9 | 63.0 | 67.4 | 65.3 | 65.6 | 68.4 | 70.8 | **71.9** |
>
>
> >**Q2:** The performance gains compared to baseline methods (in Tables 1 and 2) are indeed surprising, and it would be better to visualize the learned embeddings on both base and novel datasets to see the effects of RFD on feature learning.
>
> **R:** Thanks for the kind suggestion. We have included the t-SNE visualizations of the learned embeddings on both base and novel datasets in the [supplementary material](https://openreview.net/attachment?id=zYXFMeHRtO&name=supplementary_material). Interestingly, we observe that despite the superior performance of our method compared to the VCLIP model (or superior performance of VCLIP over frozen CLIP), there does not appear to be much visual discrepancy in the t-SNE projection of different methods. The possible reason is that the fine-grained spatial-temporal information that is essential for discriminating actions may not be well preserved in the low-dimensional space of t-SNE.
>
> However, from the visualizations of spatial-temporal attention maps (refer to Fig. 4 in our paper and Fig.6, Fig. 7, Fig. 8, and Fig. 9 in the Appendix), our method can better focus on parts related to informative patterns and actions compared to frozen CLIP and VCLIP.

---

> ### Author Response · Authors · 2023-11-20
> **Response to Reviewer FMt3 (2/2)**
>
> >**Q3:** The results in Table 1 are a little bit confusing, as the "Top1 accuracy" is only 77.8 on base categories. It would be better to further clarify the training and evaluation protocol, and the number of clips/crops used.
>
> **R:** Under the base-to-novel evaluation setting, each class consists of only 16 samples in the training set. This setting was first introduced by ViFi-CLIP [1], where the model is trained on base (seen) classes in a few-shot manner and evaluated on a set of novel (unseen) classes. Due to the limited training samples, the performance on the base categories is relatively low.
>
> For the training and evaluation protocol, we use videos with 8 frames for training and sample 3 clips (8 frames each) with 1 crop (3x1 views) for evaluation.
>
> The above protocol details have been given in Appendix A.1 and A.2 of the paper, please refer to them.
>
> >**Q4:** As the paper employs rephrased action descriptions, instead of action names, it would be interesting to show the video-text retrieval results.
>
> **R:** We appreciate this suggestion and conducted the video-text retrieval experiment. We followed the evaluation setting in Open-VCLIP [3], where the model is initially trained on the Kinetics-400 dataset and then evaluated on the video-text retrieval dataset, MSR-VTT [4], to evaluate the generalizability. Compared to Open-VCLIP, our method consistently achieves better performance in all cases.
>
> **Table 3: Video-text retrieval performance on MSR-VTT.**
> | METHOD | T2VR1 | T2VR5 | T2VR10 | V2TR1 | V2TR5 | V2TR10 |
> | :---: | :---: | :---: | :---: | :---: | :---: | :---: |
> | CLIP [2] | 31.1 | 54.2 | 63.8 | 28.9 | 53.0 | 64.9 |
> | Open-VCLIP [3] | 33.2 | 57.1 | 67.4 | 34.4 | 59.8 | 71.2 |
> | Ours | **34.6** | **57.5** | **68.0** | **35.4** | **60.3** | **71.4** |
>
> Thanks again for your valuable comments and looking forward to your reply.
>
> [1] Rasheed, Hanoona, et al. "Fine-tuned clip models are efficient video learners." Proceedings of the IEEE/CVF Conference on Computer Vision and Pattern Recognition. 2023.
>
> [2] Radford, Alec, et al. "Learning transferable visual models from natural language supervision." International conference on machine learning. PMLR, 2021.
>
> [3] Weng, Zejia, et al. "Transforming CLIP to an Open-vocabulary Video Model via Interpolated Weight Optimization." arXiv preprint arXiv:2302.00624 (2023).
>
> [4] Xu, Jun, et al. "Msr-vtt: A large video description dataset for bridging video and language." Proceedings of the IEEE Conference on Computer Vision and Pattern Recognition. 2016.

---

> > ### Comment · Reviewer_FMt3 · 2023-11-22
> > **Response to the authors' rebuttal**
> >
> > I would like to thank all the authors for your effort in this rebuttal. Your responses, especially the additional K400 performance in Table 1, and the video-text retrieval results on MSR-VTT, have addressed most of my concerns, and it is encouraging to see even if your ft results are slightly worse than that of AIM and ViFi-CLIP, the cross-dataset performance is remarkably better. I will keep my positive rating.

---

> > > ### Author Response · Authors · 2023-11-22
> > > **Reply to the feedback from Reviewer FMt3**
> > >
> > > Thanks for expressing your appreciation for the additional performance reports and giving a positive assessment of this work. It means a lot to us.

---

### Official Review · Reviewer_zWh1 · 2023-11-05

**Soundness:** 3 good
**Presentation:** 3 good
**Contribution:** 3 good
**Rating:** 6
**Confidence:** 5

**Summary:**

This paper proposed FROSTER as an open-vocabulary action recognition framework. Due to the gap between image and video domains, it is difficult to fine-tune CLIP model on video datasets without hurting its generalization ability. To this end, FROSTER employs the knowledge distillation strategy by treating the frozen CLIP as a teacher. The proposed method is evaluated on both base-to-novel and cross-dataset settings, and achieves better accuracy than existing methods without knowledge distillation on all datasets.

**Strengths:**

1. This paper has a good motivation to explore how a good open-vocabulary classifier transfers CLIP to the video domain, which is clearly shown in Figure 1.
2. The experiment is conducted comprehensively on 5 popular video datasets in terms of different network architectures.
3. The proposed method is practically useful and should be not difficult to implement.
4. The authors promise to release the source code.

**Weaknesses:**

1. I have concerns on the novelty. Knowledge distillation on CLIP is not new in this field. The authors didn't discuss related references as follows. Besides, the authors almost follow the classic knowledge distillation framework except for residual feature distillation.
[a] CLIPPING: Distilling CLIP-Based Models with a Student Base for Video-Language Retrieval, CVPR 2023.
[b] Enabling Multimodal Generation on CLIP via Vision-Language Knowledge Distillation, ACL 2022.
2. The authors didn't employ compact student model as in normal knowledge distillation. I am wondering if the performance is better if we use ViT-B/32 as teacher and ViT-B/16 as student.
3. In Fig. 1, why CLIP performs more stable on larger Kinetics dataset than HMDB51 or UCF101?
4. What is the contribution of text rephrasing in Section 4.2?

**Questions:**

Please see the weaknesses. The authors should clarify the difference from the previous knowledge distillation works and add the corresponding subsection in the related work.

---

> ### Author Response · Authors · 2023-11-20
> **Response to Reviewer zWh1 (1/2)**
>
> Thanks for your time and efforts in reviewing this paper and your detailed and insightful feedback. Your constructive suggestions are of great help to us. We are pleased that our paper is found to be well-motivated, the proposed method is practically useful and easy to implement, and the experiments are comprehensive.
>
> Please refer to our detailed response below, regarding the concerns about novelty, experiments of knowledge distillation using a small student model, and additional explanation of experimental results.
>
> >**Q1:** I have concerns on the novelty. Knowledge distillation on CLIP is not new in this field. The authors didn't discuss related references as follows. Besides, the authors almost follow the classic knowledge distillation framework except for residual feature distillation. [a] CLIPPING: Distilling CLIP-Based Models with a Student Base for Video-Language Retrieval, CVPR 2023. [b] Enabling Multimodal Generation on CLIP via Vision-Language Knowledge Distillation, ACL 2022.
>
> **R:** Thanks for pointing out the CLIP-based distillation method (CLIPPLING [1] and VLKD [2]), which helps improve the literature analysis in our paper. Below, we compare our method with CLIPPLING [1] and VLKD [2]:
>
> - **(a)** CLIPPING focuses on model compression, which aims to fully transfer the knowledge of the Clip4clip (a big teacher model) to the MobileViT-v2 (a small student model). In contrast, we aim to efficiently adopt a pre-trained CLIP model, such that the resulting model maintains the strong generalization capacity of a pre-trained CLIP while being video-specific. **(b)** The key technical contributions of CLIPPING are a layer-wise alignment scheme and a distillation strategy based on video-text distributions. Both the two components are to ensure the student model can fully absorb the knowledge of the CLIP model. Differently, we design a residual feature distillation module emphasizing the balance between generalizability maintenance and video-specific feature learning, which goes beyond just mimicking the CLIP model. **\(c\)** Besides, CLIPPING is only designed for a specific student model (MobileViT-v2), while our method can be widely used for student models with different architectures (refer to Tab. 3 of our paper).
>
> - **(a)** VLKD concentrates on aligning the features of the language model (BART) to the CLIP model and subsequently integrating them to be a multi-modal generator. In contrast, our method is more than feature alignment. Since the tuned model in our framework is well-aligned with the CLIP model at the beginning of fine-tuning, our goal is to maintain its generalizability while learning video-specific knowledge. **(b)** The key technical contribution of VLKD is proposing to align the features of BART to CLIP using feature distance minimization and contrastive loss. On the contrary, we propose to balance feature learning in two different objectives (generalizability and video-specific learning) by a residual feature distillation module.
>
> To demonstrate the superiority of our method, we evaluate the effectiveness of the VLKD method for knowledge distillation in our task. After carefully reading the CLIPPING paper, we found that its design choice is highly optimized for the specific MobileViT-V2 model in the conventional knowledge distillation setting, making it non-trivial to be modified to handle the open-vocabulary action recognition task. In contrast, our method can be easily integrated with different methods to improve their performance on open-vocabulary action recognition (please refer to Tab. 3 in the main paper).
>
> As shown in Tab. 1 below, our method achieves superior performance than the VLKD method.
>
> **Table 1: Performance (top1-acc (\%)) of different distillation methods under the cross-dataset evaluation setting.**
> | Distillation Method | UCF101 | HMDB51 | K-600 |
> | :------: | :------: | :------: | :------: |
> | VLKD | 82.6 | 52.3 | 71.8 $\pm$ 1.2 |
> | **Ours** | **85.0** | **54.5** | **74.8** $\pm$ 0.9 |
>
> We have included the above analysis in the section of related work of our revision.
>
> Besides, we agree that knowledge distillation has been a widely studied topic in the field, which is part of the motivation of our method to study open-vocabulary action recognition from a knowledge distillation perspective. However, we would like to highlight that conventional knowledge distillation methods aim to distill the same knowledge from one model to another, while differently, in our case, we have two objectives, one is maintaining the generalization capability of a pre-trained CLIP and the other is effectively adapting from image to video task. Simply applying the conventional distillation methods is not sufficient for our tasks (refer to Tab. 4 of our paper), while our method achieves superior results.

---

> ### Author Response · Authors · 2023-11-20
> **Response to Reviewer zWh1 (2/2)**
>
> >**Q2:** The authors didn't employ a compact student model as in normal knowledge distillation. I am wondering if the performance is better if we use ViT-B/32 as teacher and ViT-B/16 as student.
>
> **R:** Thanks for the suggestion. We suppose the reviewer meant using ViT-B/16 as the teacher and ViT-B/32 as the student, because '/32' and '/16' indicate the patch size for patch embedding, making ViT-B/32 a more compact model than ViT-B/16. We conduct the comparison in Tab. 1 below. We observe that using a compact student model can also consistently improve performance, which further demonstrates the effectiveness of our proposed method.
>
> **Table 1: Performance (top1-acc (\%)) of using different models for distillation under the cross-dataset evaluation setting. --- indicates only conducting model finetuning but not involving knowledge distillation.**
>
> | Student | Teacher | UCF101 | HMDB51 | K-600 |
> | :------: | :------: | :------: | :------: | :------: |
> | ViT-B/32 | --- | 78.2 | 46.8 | 62.3 $\pm$ 0.9 |
> | ViT-B/32 | ViT-B/32 | 80.0 ($\uparrow$ **1.8**) | 51.6 ($\uparrow$ **4.8**) | 68.3 $\pm$ 0.5 ($\uparrow$ **6.0**) |
> | ViT-B/32 | ViT-B/16 | 81.0 ($\uparrow$ **2.8**) | 51.2($\uparrow$ **4.4**) | 69.5 $\pm$ 0.8 ($\uparrow$ **7.2**) |
>
> >**Q3:** In Fig. 1, why CLIP performs more stable on larger Kinetics dataset than HMDB51 or UCF101?
>
> **R:** We are not completely sure what the reviewer means by "stable" here. We suspect that it refers to the fact that the frozen CLIP achieves better performance than the fine-tuned methods on the Kinetics-600 dataset, instead of the HMDB51 and UCF101 datasets. We clarify it as follows.
>
> The experiments in Fig. 1 of the paper were conducted under a cross-dataset evaluation setting, where the models were fine-tuned (except for the frozen CLIP) on the Kinetics-400 dataset and then evaluated on the UCF101, HMDB51, and Kinetics-600 datasets. Please note that in the Kinetics-600 dataset, we excluded the classes that are identical to those in the Kinetics-400 dataset. In this way, the average text-feature similarity (see Fig. 1 of our paper) of all categories between Kinetics-600 and Kinetics-400 (0.721) is smaller than that between UCF101 and Kinetics-400 (0.805) and that between HMDB51 and Kinetics-400 (0.771). Therefore, compared to the UCF101 and HMDB51 datasets, recognition performance on the Kinetics-600 dataset relies more on generalizability. Since the frozen CLIP has not been fine-tuned on video data, its pre-trained generalizability is better maintained than the fine-tuned methods. Hence, the frozen CLIP achieves better performance than fine-tuned methods on the Kinetics-600 dataset, instead of the HMDB51 and UCF101 datasets.
>
> >**Q4:** What is the contribution of text rephrasing in Section 4.2?
>
> **R:** In the last paragraph of Sec. 3.1.1 in the paper, we mention that the text rephrasing is introduced to rephrase the action names into descriptions for enriching the text diversity, which serves as text data augmentation during training, due to the limited template text provided.
>
> The impact of text rephrasing is evaluated in the last two rows of Tab. 4 of the paper, which is shown below.
>
> **Table 2: Impacts of text rephrasing under the cross-dataset evaluation setting.**
> | Method | UCF101 | HMDB51 | K-600 |
> | :------: | :------: | :------: | :------: |
> | Ours w/o text rephrasing | 84.4 | 54.4 | 73.0 $\pm$ 0.7 |
> | Ours | **85.0** | **54.5** | **74.8** $\pm$ 0.9 |
>
> We have added the above description in 3.1.1 in the paper to further clarify.
>
> Thanks again for your valuable comments and looking forward to your reply.
>
> [1] Pei, Renjing, et al. "CLIPPING: Distilling CLIP-Based Models with a Student Base for Video-Language Retrieval." Proceedings of the IEEE/CVF Conference on Computer Vision and Pattern Recognition. 2023.
>
> [2]Dai, Wenliang, et al. "Enabling multimodal generation on CLIP via vision-language knowledge distillation." Findings of Association for Computational Linguistics 2022: 2383-2395.

---

> ### Author Response · Authors · 2023-11-22
> **A Kind Reminder for Reading the Response**
>
> Dear Reviewer zWh1,
>
> Thanks for your valuable time and efforts in reviewing our paper. We understand that you may be rather busy during this period. However, as the discussion period will end in less than one day, we would like to kindly request your feedback on our responses. We would be happy to discuss with you in detail if you have additional comments about our paper.
>
> Look forward to your reply.
>
> Best regards,
>
> Paper1809 Authors

---

### Official Review · Reviewer_ZdtY · 2023-11-07

**Soundness:** 3 good
**Presentation:** 4 excellent
**Contribution:** 3 good
**Rating:** 6
**Confidence:** 4

**Summary:**

The paper addressed the problem of adapting image-based CLIP models to open vocabulary action recognition in videos. The starting point of this paper is an observation that adapting CLIP on a particular video dataset will likely reduce its performance on other distinct datasets. Based on this observation, the authors proposed a method that combines a variant of feature distillation (Residual Feature Distillation) with standard cross entropy loss for adapting CLIP to videos. The proposed method can be attached to existing adaptation methods, and demonstrated strong empirical results on public benchmarks.

**Strengths:**

* The paper addressed an important and trendy topic on adapting CLIP models for video recognition tasks.

* The paper is fairly well written. Technical details are clearly described and easy to follow.

* The proposed method is well motivated, and offers a reasonable solution.

* The experiments are extensive. The results are solid.

**Weaknesses:**

My main concern lies in the technical innovation of the paper. The proposed feature distillation falls into the framework of knowledge distillation, which has been extensively explored for adapting pre-trained models. While I agree that many prior works are insufficient for the objective of this paper, there are a few very recent papers that share the key idea of using feature distillation to adapt CLIP models for video tasks (see e.g., [a]). With this prior work, the technical innovation of the paper seems a bit incremental.

[a] Pei et al., CLIPPING: Distilling CLIP-Based Models with a Student Base for Video-Language Retrieval, CVPR 2023.

**Questions:**

I am a bit confused by how the results were reported for the proposed method in Table 1. Sec 4.2 (implementation details) states that “Otherwise stated, we use VCLIP for conducting experiments.” Is the proposed method (FROSTER) also built on VCLIP in Table 1? If so, VCLIP should be included as a baseline here. If not, a description should be included.

---

> ### Author Response · Authors · 2023-11-20
> **Response to Reviewer ZdtY (1/2)**
>
> Thanks for your time and efforts in reviewing our paper. We are delighted that our paper is found to be well-written, well-motivated, and presented a reasonable solution, along with extensive experiments.
>
> Please find the response below addressing the novelty concern related to previous works, as well as additional experimental results provided in Tab. 1.
>
> >**Q1:** The proposed feature distillation falls into the framework of knowledge distillation, which has been extensively explored for adapting pre-trained models. While I agree that many prior works are insufficient for the objective of this paper, there are a few very recent papers that share the key idea of using feature distillation to adapt CLIP models for video tasks (see e.g., CLIPPING). With this prior work, the technical innovation of the paper seems a bit incremental.
>
> **R:** Thanks for pointing out the CLIP-based distillation method (CLIPPING [1]), which helps us improve the reference comparison in our paper. Below, we compare our method with CLIPPING [1] and an additional CLIP-based distillation method, i.e., VLKD [2]:
>
> - **(a)** CLIPPING focuses on model compression, which aims to fully transfer the knowledge of the Clip4clip (a big teacher model) to the MobileViT-v2 (a small student model). In contrast, we aim to efficiently adopt a pre-trained CLIP model, such that the resulting model maintains the strong generalization capacity of a pre-trained CLIP while being video-specific. **(b)** The key technical contributions of CLIPPING are a layer-wise alignment scheme and a distillation strategy based on video-text distributions. Both the two components are to ensure the student model can fully absorb the knowledge of the CLIP model. Differently, we design a residual feature distillation module emphasizing the balance between generalizability maintenance and video-specific feature learning, which goes beyond just mimicking the CLIP model. **\(c\)** Besides, CLIPPING is only designed for a specific student model (MobileViT-v2), while our method can be widely used for student models with different architectures (refer to Tab. 3 of our paper).
>
> - **(a)** VLKD concentrates on aligning the features of the language model (BART) to the CLIP model and subsequently integrating them to be a multi-modal generator. In contrast, our method is more than feature alignment. Since the tuned model in our framework is well-aligned with the CLIP model at the beginning of fine-tuning, our goal is to maintain its generalizability while learning video-specific knowledge. **(b)** The key technical contribution of VLKD is proposing to align the features of BART to CLIP using feature distance minimization and contrastive loss. On the contrary, we propose to balance feature learning in two different objectives (generalizability and video-specific learning) by a residual feature distillation module.
>
> To demonstrate the superiority of our method, we evaluate the effectiveness of the VLKD method for knowledge distillation in our task. After carefully reading the CLIPPING paper, we found that its design choice is highly optimized for the specific MobileViT-V2 model in the conventional knowledge distillation setting, making it non-trivial to be modified to handle the open-vocabulary action recognition task. In contrast, our method can be easily integrated with different methods to improve their performance on open-vocabulary action recognition (please refer to Tab. 3 in the main paper).
>
> As shown in Tab. 1 below, our method achieves superior performance than the VLKD method.
>
> **Table 1: Performance (top1-acc (\%)) of different distillation methods under the cross-dataset evaluation setting.**
> | Distillation Method | UCF101 | HMDB51 | K-600 |
> | :------: | :------: | :------: | :------: |
> | VLKD | 82.6 | 52.3 | 71.8 $\pm$ 1.2 |
> | **Ours** | **85.0** | **54.5** | **74.8** $\pm$ 0.9 |
>
> We have included the above analysis in the section of related work of our revision.

---

> ### Author Response · Authors · 2023-11-20
> **Response to Reviewer ZdtY (2/2)**
>
> >**Q2:** I am a bit confused by how the results were reported for the proposed method in Table 1. Sec 4.2 (implementation details) states that “Otherwise stated, we use VCLIP for conducting experiments.” Is the proposed method (FROSTER) also built on VCLIP in Table 1? If so, VCLIP should be included as a baseline here. If not, a description should be included.
>
> **R:** Thanks for the suggestion. We have included the results of VCLIP in Tab. 1 of the revised manuscript. Below, we present the results, which demonstrate that our method can improve the performances for both the base and novel categories for all datasets.
>
> **Table 2: Performance (top1-acc (\%)) under the base-to-novel evaluation setting. B, N, and HM denote the base set, novel set, and harmonic mean, respectively.**
>
> | Method | K400(B) | K400(N) | K400(HM) | UCF(B) | UCF(N) | UCF(HM) | HMDB(B) | HMDB(N) | HMDB(HM) | SSv2(B) | SSv2(N) | SSv2(HM) |
> | :------: | :------: | :------: | :------: | :------: | :------: | :------: | :------: | :------: | :------: | :------: | :------: | :------: |
> | VCLIP | 76.5 | 62.6 | 68.9 | 70.3 | 50.4 | 58.7 | 94.8 | 77.5 | 85.3 | 16.0 | 11.0 | 13.0 |
> | **VCLIP w/ours** | **77.8** | **64.3** | **70.4** | **74.1** | **58.0** | **65.1** | **95.3** | **80.0** | **87.0** | **18.3** | **12.2** | **14.6** |
>
> Thanks again for your valuable comments and looking forward to your reply.
>
> [1] Pei, Renjing, et al. "CLIPPING: Distilling CLIP-Based Models with a Student Base for Video-Language Retrieval." Proceedings of the IEEE/CVF Conference on Computer Vision and Pattern Recognition. 2023.
>
> [2]Dai, Wenliang, et al. "Enabling multimodal generation on CLIP via vision-language knowledge distillation." Findings of Association for Computational Linguistics 2022: 2383-2395.

---

> ### Author Response · Authors · 2023-11-22
> **A Kind Reminder for Reading the Response**
>
> Dear Reviewer ZdtY,
>
> Thanks again for your great efforts and insightful comments in reviewing this paper! With the discussion period drawing to a close, we expect your feedback and thoughts on our reply.  We look forward to hearing from you, and we can further address unclear explanations and remaining concerns if any.
>
> Regards,
>
> Authors

---

> > ### Comment · Reviewer_ZdtY · 2023-11-22
> >
> > I appreciate the authors' effort in responding to my questions. The response provided additional results to strengthen the baselines in Table 1 and clarify the technical contribution w.r.t. prior works (e.g., CLIPPING and VLKD). My previous question about Table 1 has been fully addressed.
> >
> > However, I am still a bit concerned about the key technical innovations. Yes, I acknowledge that the proposed method differs from prior works (CLIPPING and VLKD) in the modeling and technical details. However, there is a fairly strong conceptual similarity, as also pointed out by zWh1. It is perhaps not surprising that the proposed solution can work, yet it is somewhat a question if the proposed work brings new knowledge in the field.
> >
> > That been said, I am inclined to maintain my previous rating.

---

> ### Author Response · Authors · 2023-11-23
> **Response to the feedback from Reviewer ZdtY**
>
> We thank the reviewer for the response and address the additional comments as follows:
>
> > Q1: There is a fairly strong conceptual similarity, as also pointed out by zWh1.
>
> **R:** The aim of previous methods is to mirror the representation capability from a pre-trained CLIP model as much as possible. However, in our work, the model needs to maintain the feature generalization capability as the pre-trained CLIP model and more importantly, also adapt to the downstream video action recognition task, which can not be achieved by previous knowledge distillation approaches. The model needs to learn to balance between these two distinct objectives. For instance, if we enforce the features to be overly close to the frozen CLIP features, it may hinder the video-specific learning to fit the video data. Conversely, if we overemphasize video-specific learning, the generalizability in the tuned model might be lost. Our residual feature distillation method is specifically developed to achieve both goals simultaneously, which has not been well investigated in the field.
>
>
> > Q2: It is perhaps not surprising that the proposed solution can work, yet it is somewhat a question if the proposed work brings new knowledge to the field.
>
> **R:** It is also important to note that for the task of open-vocabulary action recognition, knowledge distillation from a pre-trained CLIP model is not enough as the model has never been trained to adapt to this task (see Fig. 1 of our paper), while directly finetuning will hurt the generalization capability of the pre-trained CLIP (see Fig. 1 of our paper). Therefore, our method addresses this problem and presents a simple but effective approach to it, achieving new state-of-the-art performance on all five popular video benchmarks, which we believe can be a very helpful reference for future research in this direction.
>
> Thanks for your reading and hope you can take the above information into consideration for assessment.

---

### Author Response · Authors · 2023-11-20
**General Response**

We sincerely thank all the reviewers for their time and efforts in reviewing our paper. We are delighted that the reviewers find the paper to be well-written/well-motivated (Reviewers ZdtY, zWh1, and FMt3), the method to be interesting/simple yet effective (Reviewers FMt3 and RLyA), and the experiments to be promising/comprehensive (All Reviewers).

We have provided experiments, analyses, and clarifications to respond to each reviewer’s comments, respectively. We hope that our response has addressed all the concerns and we will be more than happy to address further concerns if there are any.

Below, we would like to highlight the major updates made to the paper during the discussion period.

* **Sec. 2.2**: Following the question raised by Reviewer ZdtY and zWh1, we have included the reference discussion in related work to clarify the differences between our method and previous CLIP-based distillation methods.

* **Tab. 1 and Sec. 4.2**: Following the kind suggestion proposed by Reviewer ZdtY and RLyA, we have included the baseline results of VCLIP in Tab. 1 to better evaluate the effectiveness of our method. Besides, the corresponding experimental analysis has been added in Sec. 4.2.

* **App. A.3**: Following the kind suggestion proposed by Reviewer FMt3, we have included the closed-set evaluation result on Kinetics-400 and provided a corresponding analysis in the App. A.3.

* **Suppl. Mat.**: Following the kind suggestion proposed by Reviewer FMt3, we have included visualizations of learned embeddings from both base and novel datasets, and provided corresponding analysis in the [supplementary material](https://openreview.net/attachment?id=zYXFMeHRtO&name=supplementary_material).

---

### Author Response · Authors · 2023-11-23
**To All Reviewers and Area Chairs**

Dear reviewers and area chairs,

We sincerely thank all reviewers and area chairs for their valuable time and comments. During the discussion, we responded to all reviewers’ comments and provided experiments/analyses/clarifications to solve their concerns. Here, we summarize why our method is different from previous distillation methods from the perspectives of objectives, technical innovations, and experimental results:

- The aim of previous methods is to mirror the representation capability from a pre-trained CLIP model as much as possible. However, in our work, the model needs to maintain the feature generalization capability as the pre-trained CLIP model and more importantly, also adapt to the downstream video action recognition task, **which can not be achieved by previous knowledge distillation approaches**. The model needs to learn to balance between these two distinct objectives, which are necessary and non-trivial. **Our residual feature distillation method is specifically developed to achieve both goals simultaneously, which has not been well investigated in the field.** Please refer to [Response to the feedback from Reviewer ZdtY](https://openreview.net/forum?id=zYXFMeHRtO&noteId=ei7q4kfuNG) or [Response to Reviewer ZdtY (1/2)](https://openreview.net/forum?id=zYXFMeHRtO&noteId=jTQKcnnuex) for more details.
- The technical designs in previous methods are to ensure the student model can fully absorb the knowledge of the CLIP model. Differently, we design a residual feature distillation module emphasizing the balance between generalizability maintenance and video-specific feature learning. Please refer to [Response to Reviewer ZdtY (1/2)](https://openreview.net/forum?id=zYXFMeHRtO&noteId=jTQKcnnuex) or [Response to Reviewer zWh1 (1/2)](https://openreview.net/forum?id=zYXFMeHRtO&noteId=7zwCiJ5aYW) for more details.
- In experiments, we also demonstrate that the proposed residual feature distillation module outperforms both the classical distillation approaches (refer to Tab. 4 of our paper) and [recent CLIP-based distillation methods](https://openreview.net/forum?id=zYXFMeHRtO&noteId=jTQKcnnuex). **These experiments prove that previous methods are insufficient for our task, while our method is superior.**

Thanks again for the efforts of all reviewers and area chairs.

Best,

Paper1809 Authors

---

### Meta-Review · Area_Chair_Yaa7 · 2023-12-17

**Metareview:**

This paper adapted image-based CLIP models to open vocabulary action recognition in video via knowledge distillation. All reviewers give positive comments by acknowledging the efforts on newly added experiments and explanations after rebuttal. The AC thus decided to accept it.

**Justification For Why Not Higher Score:**

This paper proposed a new residual feature distillation module, which is useful but not a breakthrough in the field.

**Justification For Why Not Lower Score:**

The experiments are good and meaningful for open-vocabulary action recogniton field.

---

### Decision · Program_Chairs · 2024-01-16

Accept (poster)